# Ultrasonographic Evaluation of the Shoulders and Its Associations with Shoulder Pain, Age, and Swim Training in Masters Swimmers

**DOI:** 10.3390/medicina57010029

**Published:** 2020-12-31

**Authors:** Yuta Suzuki, Noriaki Maeda, Junpei Sasadai, Kazuki Kaneda, Taizan Shirakawa, Yukio Urabe

**Affiliations:** 1Department of Sports Rehabilitation, Graduate School of Biomedical and Health Sciences, Hiroshima University, Hiroshima 734-8553, Japan; yt.suzuki28@gmail.com (Y.S.); norimmi@hiroshima-u.ac.jp (N.M.); kzknswp@gmail.com (K.K.); 2Department of Rehabilitation, Matterhorn Rehabilitation Hospital, Hiroshima 737-0046, Japan; 3Sports Medical Center, Japan Institute of Sports Sciences, Tokyo 115-0056, Japan; jumpei.sasadai@jpnsport.go.jp; 4Department of Rehabilitation, Koyo Orthopedic Clinic, Hiroshima 739-1733, Japan; 5Department of Orthopedics, Matterhorn Rehabilitation Hospital, Hiroshima 737-0046, Japan; matter@jasmine.ocn.ne.jp

**Keywords:** swimming, masters, ultrasound, shoulder pain, rotator cuff, tendinosis, calcification

## Abstract

*Background and objectives:* The long head of the biceps (LHB) and rotator cuff tendinopathy is the major cause of shoulder pain in competitive swimmers. The risk of tendinopathy increases with aging; however, the structural changes of LHB and rotator cuff in populations of masters swimmers have not been well examined. The purpose of this study was to investigate the prevalence of ultrasonographic abnormalities of the shoulders in masters swimmers, and the association of pain, age, and swim training with structural changes in this population. *Materials and Methods:* A total of 60 subjects participated in this study, with 20 masters swimmers with shoulder pain, 20 asymptomatic masters swimmers, and 20 sex- and age-matched controls. All swimmers completed a self-reported questionnaire for shoulder pain, their history of competition, and training volume. Each subject underwent ultrasonographic examination of both shoulders for pathologic findings in the LHB tendon, rotator cuff (supraspinatus (SSP) and subscapularis (SSC)) tendons, and subacromial bursa (SAB) of both shoulders and had thickness measured. *Results:* The prevalence of tendinosis (LHB, 48.8%; SSP, 17.5%; SSC, 15.9%), partial tear (SSP, 35.0%), and calcification (SSC, 10.0%) were higher in swimmers than in controls. LHB and SSP tendinosis were associated with shoulder pain. Older age and later start of competition were associated with an increased risk of LHB tendinosis and SSC calcification. Earlier initiation of swimming and longer history of competition were associated with an increased risk of SSP and SSC tendinosis. The thicker SSP tendon significantly increased the risk of tendinosis and partial tear. *Conclusions:* A high prevalence of structural changes in the rotator cuff and biceps tendons in masters swimmers reflects the effect of shoulder symptoms, aging, and swim training.

## 1. Introduction

Swimming is a popular recreational and competitive sport that is popular with all generations from children to the elderly (aged 65 years or older). Many swimmers return to competition at an older age for health benefits. Indeed, over 38,000 masters swimmers compete in Japan.

Shoulder pain in swimmers is a common orthopedic symptom that is termed the “swimmer’s shoulder.” The incidence of shoulder pain is reported to be 19–62% in masters swimmers [1,2,3]. Although several studies reported the potential risk factors for shoulder pain, including intrinsic (shoulder instability, range of motion, rotator cuff muscle strength, and scapular dyskinesis) and extrinsic factors (age, competitive level, years of competition, training volume, and swim training equipment) [4,5], none of the risk factors identified as a high level of certainty [5]. Thus, these various factors are likely to interact to cause shoulder pain.

The magnetic resonance imaging (MRI) investigation revealed that the most common structural change in the swimmer’s painful shoulder is supraspinatus (SSP) tendinopathy, which is the direct cause of shoulder pain [6]. In recent years, such structural changes of the rotator cuff in swimmers have been evaluated by ultrasound as an accessible, portable, valid, and reliable tool [7,8,9,10,11]. These previous studies show that SSP tendinosis is identified in 44% of collegiate swimmers and 96% of Olympic swimmers [7,8]. Higher SSP tendon thickness is correlated with SSP tendinosis and is also associated with shoulder pain [7]. The prevalence of these structural changes increases with the number of years of competition as well as with the training volume [7,8].

Despite these findings, the structural alterations of shoulders in competitive masters swimmers have not been well examined. Since increasing age is a specific risk factor for rotator cuff abnormalities [12,13,14], it is likely that the risk of developing a “swimmer’s shoulder” increases in masters swimmers due to repetitive swimming activities and the effects of aging. Furthermore, the relationships between structural changes in the shoulders and symptoms and the swim training volume in such populations are not well understood.

Given high shoulder injury rates in swimming, the characteristics of structural changes in the rotator cuff and the identification of these associated factors may provide important knowledge to establish the prevention and treatment of shoulder pain in masters swimmers.

The purpose of this study was to investigate the prevalence of ultrasonographic abnormalities of the shoulders in competitive masters swimmers and to assess the extent to which shoulder pain, age, years of competition, and swim training volume are associated with structural changes in tendons and the bursa in this population. The present study hypothesized that swimmers would demonstrate higher prevalence of rotator cuff tendinopathy than healthy controls and that SSP tendinosis would be associated with shoulder pain and the number of years of competition.

## 2. Materials and Methods

### 2.1. Participants

A priori power analysis was performed using G*Power 3.1 (Kiel University, Germany). To detect a statistical difference among healthy adults and masters swimmers with and without shoulder pain, the chi-squared test determined that 13 subjects in each group would be necessary with an estimated effect size of power set at 0.80 and the significance set at α = 0.05. In addition, based on preliminary studies [15], to detect a difference in tendon thickness > 1 mm with a standard deviation of 1 mm among each group with a power set at 0.95 and significance set at α = 0.05, power ANOVA indicated a sample size of at least n = 14 per group.

A total of 60 subjects (23 men and 37 women) aged 33 to 65 years participated in this study; 40 were competitive swimmers and 20 were healthy sex- and age-matched controls. The common inclusion criteria were as follows: aged 30 to 65 years at the time of the survey and no history of shoulder fractures, shoulder dislocation, or shoulder surgery. In addition, swimmers included those who had practiced swimming regularly at least once a week in the preceding year, while controls were those who were not engaged in regular exercise or sports activities at enrolment. We excluded swimmers with less than three years of competitive swimming history and controls with shoulder pain in ADL at the time of the survey. All participants signed an informed consent form approved by the Ethics Committee for Epidemiology of Hiroshima University (approved ID: E-2064).

### 2.2. Questionnaire Survey

Demographic information such as age, sex, height, weight, and arm dominance were recorded. In addition, each swimmer completed a questionnaire about their history of swimming competition, training volume, history of shoulder pain or injury during swimming competition (“Do you currently have any shoulder pain while swimming?”), duration of shoulder pain (“How long have you had shoulder pain while swimming?”), and appearance phase of shoulder pain (“When does the shoulder pain appear?”). All swimmers were in regular and continuous swim training with or without shoulder pain at the time of the study.

### 2.3. Ultrasonographic Evaluation

All ultrasonographic evaluations were performed by a physical therapist with more than 5 years of musculoskeletal scanning experience using a SONIMAGE HS1 with a 4–18 MHz linear probe (Konica Minolta Inc., Tokyo, Japan) according to the technical guidelines of the Ultrasound Subcommittee of the European Society of Musculoskeletal Radiology [16]. The pathologic findings of the long head of the biceps (LHB) tendon, rotator cuff (SSP and subscapularis (SSC)) tendons, and the subacromial bursa (SAB) were examined. Using the images obtained from ultrasound, the tendon and SAB thickness were measured using the Image J version 1.52q software (National Institutes of Health, Bethesda, MA, USA).

#### 2.3.1. LHB Tendon

The LHB tendon was evaluated for the presence of tendinosis and tears (partial-thickness or full-thickness). LHB tendinosis was defined as the presence of a hypoechoic defect and tendon thickening, but without defects or tendon fiber discontinuity, and an LHB tendon tear was defined as discontinuity with the absence of the LHB tendon in the bicipital groove [17,18,19]. The LHB tendon thickness was measured as the perpendicular line distance from the center of the bicipital groove at the point of the SSC muscle that appeared when the transducer was moved up and down (Figure 1) [20,21].

#### 2.3.2. SSP and SSC Tendons

The SSP tendon was assessed in a sitting position with the palm placed on the superior of the iliac wing with the shoulder extended and the elbow flexed in the neutral position [16]. The superior facet of the greater tuberosity was identified via the monitor of the ultrasonography device, and the transducer was placed parallel to the scapular spine and perpendicular to the SSP tendon. To visualize the SSC tendon, the transducer was placed in the same position to evaluate the LHB tendon and the forearm was rotated externally with the palm up and the elbow close to the lateral abdomen [16]. Rotator cuff tendons were evaluated for the presence of tendinosis, tears (partial-thickness, full-thickness), and calcification. Tendinosis was defined as tendon thickening or thinning associated with abnormal echogenicity and loss of the regular parallel structure of fibers in the tendon. A full-thickness tear was defined as discontinuity of the tendon fibers from the articular surface to the bursal surface appearing as a hypoechoic or anechoic defect. A partial-thickness tear was defined as a partial tear of the tendon fibers in either the bursal surface or the articular surface including a focal anechoic or hypoechoic defect that does not traverse the entire thickness of the tendon. Calcification was defined as intra-tendinous hyperechoic areas with or without posterior acoustic shadowing [18,19,22]. The SSP tendon thickness was measured from the top to the bottom at the first change point of the inclination angle (Figure 2) [20], while SSC tendon thickness was measured from the top to the bottom at a distance of 2 cm medial to the LHB tendon (Figure 3) [21].

#### 2.3.3. SAB

The transducer was placed in the same position as when the SSP tendon was evaluated. The SAB was evaluated for the presence of thickening/effusion and defined as focal or diffuse bursal thickening of more than 2 mm [23,24]. SAB thickness was measured from the top to the bottom at the first change point of the inclination angle, in the same manner as for SSP tendon thickness measurement (Figure 2) [20].

### 2.4. Statistical Analysis

All statistical analyses were performed using JMP^®^ Pro version 14.2 for Mac (SAS Institute Inc., Cary, NC, USA). The baseline characteristics, prevalence of ultrasonographic findings, and tendon and bursa thickness were compared among healthy masters swimmers, symptomatic masters swimmers, and the control group using a one-way ANOVA for quantitative variables and the chi-squared test for qualitative variables. When appropriate, follow-up analysis was performed using the Tukey–Kramer test or residual analysis for the post-hoc test. Effect sizes were calculated using Cohen’s d or Cramer’s V statistics, and the post-hoc observed power was generated by the G *Power software. Independent t-tests were used to compare the years of competition and training volume between masters swimmers with and without shoulder pain and the tendon and bursa thickness between the sides (dominant vs. non-dominant or affected vs. non-affected) within groups. In swimmers, logistic regression analysis was used to assess the relationship between abnormal ultrasonographic findings and pain, age, and variables of swim training. Adjusted odds ratio (OR) was reported to quantify the magnitude of the association between specific variables and abnormal ultrasonographic findings. Statistical significance was set at α = 0.05.

## 3. Results

### 3.1. Demographics of the Participants

The study group included 20 healthy individuals (8 men and 12 women; mean age: 51.2 ± 9.0 years; mean body mass index: 21.1 ± 1.9 kg/m^2^; control group), and 40 masters swimmers. The masters swimmers comprised 20 swimmers with no history of shoulder injury (8 man and 12 women; mean age: 51.8 ± 10.7 years; mean body mass index: 22.5 ± 2.3 kg/m^2^; healthy group) and 20 swimmers with current shoulder pain (7 men and 13 women; mean age: 51.4 ± 10.5 years; mean body mass index: 22.0 ± 1.9 kg/m^2^; symptomatic group). There were no significant differences in sex, age, BMI.

The swimmers had been competing for a mean of 18.3 ± 10.5 years in the healthy group and 17.4 ± 11.7 years in the symptomatic group, and the age at the start of competition was 40.1 ± 15.9 and 39.3 ± 15.2 years, respectively. The swimmers had 3.1 ± 1.9 training sessions per week in the healthy group and 3.3 ± 1.5 training sessions per week in the symptomatic group with the training volume of 2970 ± 2070 m and 2850 ± 1820 m, respectively. There were no significant differences in the history of swimming competition and training volume. All the symptomatic swimmers reported unilateral pain, and 12 swimmers (60.0%) had shoulder pain on the dominant side. The duration of shoulder pain ranged from 2 weeks to 1 year, 6 swimmers (30.0%) complained of shoulder pain not only during swim training, but also after training, and 1 swimmer (5.0%) had pain at night. The presence of shoulder pain did not affect the practice or competition of swimming.

### 3.2. Ultrasonographic Findings

Table 1 summarizes results of the comparison of ultrasonographic abnormalities between the groups and Table 2 demonstrates the laterality of ultrasonographic findings within groups.

#### 3.2.1. LHB Tendon

LHB tendinosis was found in 5/40 shoulders (12.5%) in the controls and 39/80 shoulders (48.8%) in the swimmers. The prevalence of LHB tendinosis was significantly higher in the swimmers (*p* < 0.01), and LHB tendinosis was significantly more frequent on the affected side of symptomatic swimmers. No significant difference was found between groups or within groups with regards to LHB tendon thickness (*p* > 0.05).

#### 3.2.2. SSP Tendon

SSP tendinosis was found in 7/40 shoulders (17.5%) in the controls and 60/80 shoulders (75.0%) in the swimmers. The prevalence of SSP tendinosis was significantly higher in the swimmers (*p* < 0.01). Partial SSP tendon tears were seen in 1/20 shoulders (2.5%) in controls, 5/40 shoulders (12.5%) in healthy swimmers, and 9/40 shoulders (22.5%) in symptomatic swimmers. The prevalence was highest in symptomatic swimmers, followed by healthy swimmers and controls (*p* < 0.05). SSP tendon was significantly thicker in symptomatic swimmers (7.31 ± 0.92), followed by healthy swimmers (6.80 ± 0.91) and controls (5.76 ± 0.74) (*p* < 0.01).

#### 3.2.3. SSC Tendon

SSC tendinosis was found in 6/40 shoulders (15.0%) in the controls and 38/80 shoulders (47.5%) in the swimmers, and calcification was observed in 4/40 shoulders (10.0%) in the controls and 31/80 shoulders (38.8%) in the swimmers. The prevalence of tendinosis and calcification was significantly higher in the swimmer group (*p* < 0.01). SSC tendon thickness was significantly greater in the swimmers than in the controls (*p* < 0.01), but there was no significant difference within swimmers.

#### 3.2.4. SAB

The prevalence of SAB thickening/effusion was not significantly different between the groups. The thickness of SAB was significantly greater in healthy (1.38 ± 0.34) and symptomatic swimmers (1.44 ± 0.24) than in the controls (1.05 ± 0.35) (*p* < 0.01).

### 3.3. Factors Associated with Ultrasonographic Findings

The results of the regression analysis are summarized in Table 3.

#### 3.3.1. LHB Tendinosis

If shoulder pain was present, the odds of LHB tendinosis were more than 3 times higher than in the swimmers with no history of shoulder pain (odds ratio (OR), 3.95; 95% confidence interval (CI), 1.20–15.69). Older age and later start of swimming competition were found to be associated with LHB tendinosis, with odds increasing by 5% (OR, 1.05; 95% CI, 1.00–1.11) and 4% (OR, 1.04; 95% CI, 1.00–1.08) for each year of age increase, respectively.

#### 3.3.2. SSP Tendinosis and Partial-Thickness Tear

If shoulder pain was present, the odds of SSP tendinosis were more than 4 times higher than in the swimmers with no history of shoulder pain (OR, 4.17; 95% CI, 1.60–27.92). Earlier initiation of swimming (OR, 0.95; 95% CI, 0.89–0.99) and a longer history of competition (OR, 1.13; 95% CI, 1.03–1.24) were associated with an increased risk of tendinosis. Furthermore, an increased amount of swim training per day (OR, 4.64; 95% CI, 1.20–17.98) was associated with an increased risk of tendinosis. The thicker SSP tendon significantly increased the risk of tendinosis (OR, 2.51; 95% CI, 1.05–6.03).

Only increasing age was found to be associated with SSP partial-thickness tears, with odds increasing by 11% (OR, 1.11; 95% CI, 1.02–1.22) for each year of age increase. In addition, a thicker tendon was found to significantly increase the risk of partial-thickness tears (OR, 2.42; 95% CI, 1.05–5.56).

#### 3.3.3. SSC Tendinosis and Calcification

Earlier initiation of swimming (OR, 0.96; 95% CI, 0.92–0.99) and a longer history of competition (OR, 1.06; 95% CI, 1.00–1.13) were associated with an increased risk of SSC tendinosis.

Older age and later start of swimming competition were found to be associated with SSC calcification, with odds increasing by 6% (OR, 1.06; 95% CI, 1.00–1.11) and 4% (OR, 1.04; 95% CI, 1.00–1.09) for each year of age increase, respectively.

## 4. Discussion

This study applied ultrasonography to investigate the primary structural changes of the LHB tendon, rotator cuff tendons, and SAB in masters swimmers. This is the first comprehensive study of such populations of the masters age group to explore structural shoulder abnormalities using ultrasound as well as the association between structural changes and perceived pain, age, and repetitive load of swim training.

We found increased structural changes, including partial-thickness tears in the SSP tendon, and a high prevalence of tendinopathy involving the rotator cuff and LHB tendons in masters swimmers compared to the sex- and age-matched controls. Moreover, the rotator cuff tendons and SAB were also thicker in the swimmers than in the controls. In the control group, the prevalence of ultrasonographic abnormalities was comparable and in good agreement with previous reports [22,25]. In addition, the tendon thickness values obtained from the controls were similar to a previous study for healthy volunteers, which reported a thickness of 2.1–3.2 mm for the LHB tendon, 4.5–5.6 mm for the SSP tendon, 3.8–5.1 mm for the SSC tendon, and 0.7–1.2 mm for the SAB [20,21,26]. Thus, the apparently high prevalence of intrinsic structural alternations and tendon/bursa thickening in masters swimmers likely reflects the effects of the cumulative overhead load of swimming competition.

Morphological changes consistent with tendinosis were commonly reported in the LHB and SSP tendons in younger competitive swimmers, with a prevalence of 47–72% and 44–96%, respectively [6,7,8]. In this study, we found tendinosis at a rate of 48.8% in LHB and 75.0% in SSP tendons, as in previous studies; we additionally identified SSC tendinosis with a prevalence of 47.5%. Although it is still poorly understood how tendinosis develops, it is generally accepted that excessive stress exceeds the healing capacity of tendon cells and fails to repair [27,28]. Excessive stress is the most substantial factor in the development of tendinopathy, as reflected by the fact that rotator cuff tendinopathy occurs more frequently in the dominant limb and in overhead work [14,29]. Thus, the intrinsic tendon changes of swimmers are most likely caused by overuse of the shoulders due to long-term swimming training. From the results of our regression analyses, earlier initiation and a longer history of competition increased the risk of SSP and SSC tendinosis. These results are consistent with previous research that showed an association between the presence of tendinosis and years of competition in elite swimmers [6]. Elite swimmers may swim an average of 12,000 m/day and perform more than 2500 shoulder revolutions per day [30]. In the current study, masters swimmers swam approximately 3000 m/week and performed more than 600 shoulder revolutions per week while generating propulsive force by their upper limbs in the water compared to adults of the same age. It is reasonable to assume that these repetitive shoulder revolutions quickly overload the shoulders of swimmers.

Furthermore, our results showed an association between tendinosis of LHB and SSP and shoulder pain, in agreement with previous studies in elite swimmers [6,8]. The molecular mechanisms that lead to pain in tendinosis are not well established, although the presence of inflammatory cells and their role in tendinosis have been debated [31,32]. However, it is clear that overuse triggers tendinosis, and tendinosis is associated with perceived shoulder pain, as our study and previous studies have already identified. Interestingly, although both LHB tendinosis and SSP tendinosis were found to be associated with shoulder pain, the risk factors for each were contrasting; swimmers who were older and started competition later had a higher risk of LHB tendinosis, while swimmers who had a longer history of swimming competition had a higher risk of SSP tendinosis. This suggests that the cause of shoulder pain in masters swimmers may be due to two factors: chronic SSP tendinosis caused by long-term swim training in swimmers who started competition earlier, and acute LHB tendinosis associated with a later start of competition.

Subacromial impingement during a swimming stroke is a factor of a “swimmer’s shoulder” that may be effective in explaining the structural tendon changes in swimmers. The subacromial space is narrowed around the point of hand entry because the shoulder is in forward flexion, adduction, and internal rotation: this is the typical position of subacromial impingement. Even with single swimming practice, the SSP tendon can become acutely thickened [11], and long-term swimming training causes thickening of the SSP tendon and SAB [7,9]. Thickening of the SSP tendon and SAB may relatively narrow the subacromial space and lead to an increased risk of impingement during a swimming stroke. Dischler et al. [7] reported that SSP tendon thickness over 6.2 mm increases the risk of tendinosis. In the current study, the SSP tendon was the thickest in symptomatic shoulders, and the thickness in masters swimmers met the criteria, while SAB thickening in masters swimmers is an adaptive process of swim training that is not associated with pain [9]. Thus, it is likely that acute/chronic SSP tendon thickening from single or prolonged swim training will increase the risk of impingement, while mechanical stresses on the tendon can cause morphological changes and pain.

All of the past studies on ultrasonography evaluation for swimmers have focused on the LHB, SSP, and SAB. Although the SSC has not been mentioned in previous studies, it plays a very important role in swimming and is active throughout a swimming stroke [33]. In the present study, tendinosis and calcification of the SSC were found more frequently in the swimmers than in the controls; these may reflect the effect of repetitive SSC loading during a swimming stroke. This study failed to show an association between pathological changes of the SSC and pain. However, overloading of the SSC during a swimming stroke is likely to lead to poor steering of the scapula, which affects glenohumeral joint stability [34,35] and causes secondary impingement and shoulder pain.

There are some limitations to the current study. First, the study failed to examine the history of overhead sports and occupation, which might influence rotator cuff degeneration [14]. Second, the duration of shoulder pain in the symptomatic group varied from 2 weeks to 1 year. Finally, no physical examination was performed to measure shoulder laxity, range of motion, which might also be associated with shoulder impingement and pain in swimmers [5,6].

Swimming is frequently recommended for people suffering from injuries and illnesses, especially for the elderly. However, swimming is likely to increase the risk of structural changes of the rotator cuff and biceps tendons, and aging is an unavoidable factor that causes these changes in the shoulders of masters swimmers. The structural changes of tendons in masters swimmers may coexist with the effects of aging (degeneration) and swim training (adaptation), and preventive strategies should be considered. For prevention of shoulder pain among masters swimmers, we will be required to provide (1) knowledge about the prevalence and symptoms of swimming-related shoulder pain, (2) opportunities for medical examinations for early detection of shoulder pain or structural changes that may lead to shoulder pain, and (3) construction of intervention programs for the painful shoulder of masters swimmers. For example, the effectiveness of exercises for tendinosis has already been demonstrated [36]. A longitudinal study to examine the effect of intervention on rotator cuff training or a well-balanced and progressively increasing swim training program may provide information on the management and prevention of shoulder pain among masters swimmers.

## 5. Conclusions

We found a high prevalence of structural changes in the rotator cuff and LHB tendons in competitive masters swimmers, and these findings likely coexist with the effects of both cumulative swimming activities and aging. Overuse tendinopathy of the LHB and SSP was associated with shoulder symptoms.

## Figures and Tables

**Figure 1 medicina-57-00029-f001:**
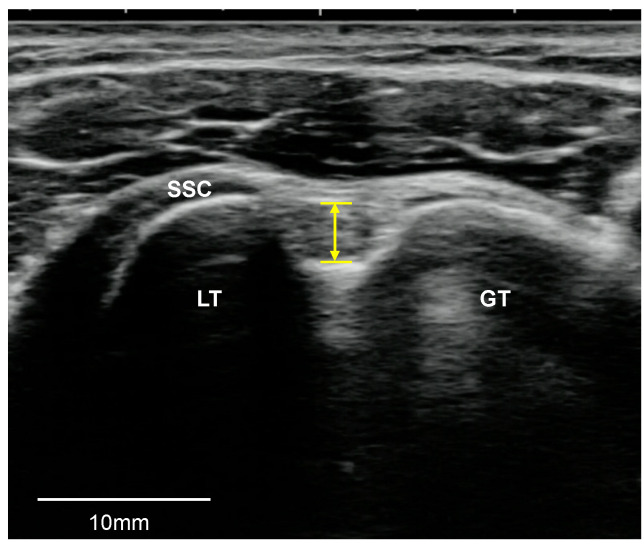
The measurement of the biceps tendon thickness. SSC, subscapularis tendon; GT, greater tuberosity; LT, lesser tuberosity.

**Figure 2 medicina-57-00029-f002:**
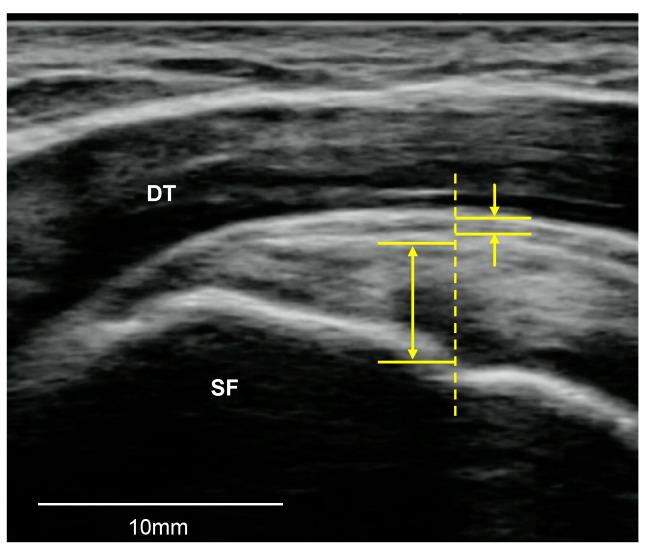
The measurement of the supraspinatus tendon and subacromial bursa thickness. SF, superior facet; DT, deltoid muscle.

**Figure 3 medicina-57-00029-f003:**
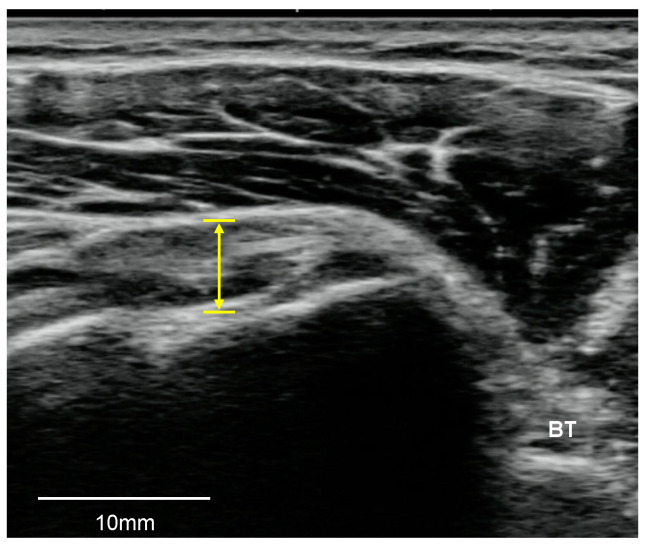
The measurement of the subscapularis tendon thickness. BT, biceps tendon.

**Table 1 medicina-57-00029-t001:** Comparison of the prevalence of ultrasonographic findings and rotator cuff thickness between healthy and symptomatic masters swimmers and the control group.

Variables	Control (*n* = 40)	Masters Swimmer		*p*-Value		Effect Size	Observed Power
		Healthy (*n* = 40)	Symptomatic (*n* = 40)	Trend	Post-hoc		
LHB tendon							
Tendinosis	5 (12.5%)	18 (45.0%)	21 (52.5%)	<0.01	Symptomatic, Healthy > Control	0.36	0.96
Partial-thickness tear	0 (0%)	1 (2.5%)	0 (0%)	n.s.	N.A.	0.13	0.23
Thickness (mm)	2.66 ± 0.49	2.94 ± 0.49	2.85 ± 0.43	n.s.	N.A.	0.11	0.14
SSP tendon							
Tendinosis	7 (17.5%)	28 (70.0%)	32 (80.0%)	<0.01	Symptomatic, Healthy > Control	0.57	0.99
Partial-thickness tear	1 (2.5%)	5 (12.5%)	9 (22.5%)	0.02	Symptomatic > Healthy > Control	0.25	0.68
Full-thickness tear	0 (0%)	1 (2.5%)	2 (5.0%)	n.s.	N.A.	0.13	1.23
Calcification	3 (7.5%)	7 (17.5%)	10 (25.0%)	n.s.	N.A.	0.19	0.45
Thickness (mm)	5.26 ± 0.74	6.80 ± 0.91	7.31 ± 0.92	<0.01	Symptomatic > Healthy > Control	0.62	0.99
SSC tendon							
Tendinosis	6 (15.0%)	20 (50.0%)	18 (45.0%)	<0.01	Symptomatic, Healthy > Control	0.32	0.89
Partial-thickness tear	0 (0%)	0 (0%)	2 (5.0%)	n.s.	N.A.	0.18	0.42
Calcification	4 (10.0%)	16 (40.0%)	15 (37.5%)	<0.01	Symptomatic, Healthy > Control	0.30	0.84
Thickness (mm)	4.76 ± 0.78	5.87 ± 1.06	6.18 ± 0.67	<0.01	Symptomatic, Healthy > Control	0.57	0.99
SAB							
Thickening/Effusion	1 (2.5%)	4 (10.0%)	5 (12.5%)	n.s.	N.A.	0.18	0.41
Thickness (mm)	1.05 ± 0.35	1.38 ± 0.34	1.44 ± 0.24	<0.01	Symptomatic, Healthy > Control	0.17	0.29

Values are presented as *n* (%) or means ± standard deviation (SD). LHB, long head of the biceps; SSP, supraspinatus; SSC, subscapularis; SAB, subacromial bursa; n.s., non-significant; N.A., not applicable.

**Table 2 medicina-57-00029-t002:** Lateral differences in the prevalence of ultrasonographic findings and rotator cuff thickness within groups.

Variables	Control		Masters Swimmers			
			Healthy		Symptomatic	
	Dominant (*n* = 20)	Non-Dominant (*n* = 20)	Dominant (*n* = 20)	Non-Dominant (*n* = 20)	Affected Side (*n* = 20)	Non-Affected Side (*n* = 20)
LHB tendon						
Tendinosis	3 (15.0%)	2 (10.0%)	7 (35.0%)	11 (55.0%)	14 (70.0%) *	7 (35.0%)
Partial-thickness tear	0 (0%)	0 (0%)	1 (5.0%)	0 (0%)	0 (0%)	0 (0%)
Thickness (mm)	2.69 ± 0.50	2.64 ± 0.50	3.01 ± 0.57	2.87 ± 0.42	2.91 ± 0.44	2.78 ± 0.44
SSP tendon						
Tendinosis	3 (15.0%)	4 (20.0%)	14 (70.0%)	14 (70.0%)	18 (90.0%)	13 (65.0%)
Partial-thickness tear	1 (5.0%)	0 (0%)	4 (20.0%)	1 (5.0%)	6 (30.0%)	3 (15.0%)
Full-thickness tear	0 (0%)	0 (0%)	1 (5.0%)	0 (0%)	1 (5.0%)	1 (5.0%)
Calcification	2 (10.0%)	1 (5.0%)	4 (20.0%)	3 (15.0%)	7 (35.0%)	3 (15.0%)
Thickness (mm)	5.35 ± 0.73	5.16 ± 0.80	6.80 ± 1.15	6.91 ± 0.68	7.54 ± 0.97 *	7.06 ± 0.86
SSC tendon						
Tendinosis	3 (15.0%)	3 (15.0%)	10 (50.0%)	10 (50.0%)	10 (50.0%)	8 (40.0%)
Partial-thickness tear	0 (0%)	0 (0%)	0 (0%)	0 (0%)	2 (10.0%)	0 (0%)
Calcification	2 (10.0%)	2 (10.0%)	9 (45.0%)	7 (35.0%)	7 (35.0%)	8 (40.0%)
Thickness (mm)	4.67 ± 0.84	4.91 ± 0.75	5.82 ± 0.97	5.91 ± 1.19	6.00 ± 0.66	6.34 ± 0.66
SAB						
Thickening/Effusion	1 (5.0%)	0 (0%)	2 (10.0%)	2 (10.0%)	2 (10.0%)	3 (15.0%)
Thickness (mm)	1.00 ± 0.41	1.10 ± 0.28	1.30 ± 0.34	1.45 ± 0.34	1.43 ± 0.28	1.45 ± 0.28

Values are presented as *n* (%) or means ± standard deviation (SD). LHB, long head of the biceps; SSP, supraspinatus; SSC, subscapularis; SAB, subacromial bursa. * Significant differences between the affected side and the non-affected side in symptomatic masters swimmers (*p* < 0.05).

**Table 3 medicina-57-00029-t003:** Results from regression analysis for factors associated with ultrasonographic outcomes.

Variables	LHB		SSP				SSC			
	Tendinosis		Tendinosis		Partial-Thickness Tear		Tendinosis		Calcification	
	OR (95% CI)	*p*-Value	OR (95% CI)	*p*-Value	OR (95% CI)	*p*-Value	OR (95% CI)	*p*-Value	OR (95% CI)	*p*-Value
Shoulder pain	3.95 (1.20–15.69)	0.02	4.17 (1.60–27.92)	0.04	3.18 (0.92–11.00)	n.s.	1.63 (0.56–4.79)	n.s.	1.09 (0.39–3.09)	n.s.
Age	1.05 (1.00–1.11)	0.03	0.97 (0.91–1.03)	n.s.	1.11 (1.02–1.22)	<0.01	0.96 (0.91–1.01)	n.s.	1.06 (1.00–1.11)	0.04
Years of competition	0.99 (0.94–1.04)	n.s.	1.13 (1.03–1.24)	<0.01	1.02 (0.96–1.08)	n.s.	1.06 (1.00–1.13)	0.03	0.99 (0.94–1.05)	n.s.
Age at start of competition	1.04 (1.00–1.08)	0.03	0.94 (0.89–0.99)	0.01	1.05 (0.99–1.10)	n.s.	0.96 (0.92–0.99)	0.02	1.04 (1.00–1.09)	0.03
Swim-training										
Frequency per week	1.13 (0.83–1.57)	n.s.	0.87 (0.60–1.27)	n.s.	1.09 (0.72–1.64)	n.s.	0.83 (0.59–1.18)	n.s.	1.06 (1.77–1.47)	n.s.
Volume per week	1.12 (0.87–1.45)	n.s.	1.38 (0.92–2.06)	n.s.	1.24 (0.92–1.68)	n.s.	0.94 (0.72–1.22)	n.s.	1.03 (0.80–1.33)	n.s.
Volume per day	0.86 (0.40–1.86)	n.s.	4.64 (1.20–17.98)	<0.01	1.47 (0.55–3.92)	n.s.	1.39 (0.63–3.06)	n.s.	0.70 (0.31–1.60)	n.s.
Tendon thickness										
LHB	1.07 (0.43–2.69)	n.s.	−		−		−		−	
SSP	−		2.51 (1.05–6.03)	0.03	2.42 (1.05–5.56)	0.02	−		−	
SSC	−		−		−		1.38 (0.77–2.47)	n.s.	0.90 (0.53–1.54)	n.s.

LHB, long head of the biceps; SSP, supraspinatus; SSC, subscapularis; OR, odds ratio; CI, confidence interval.

## Data Availability

The data presented in this study are available on request from the corresponding author.

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
