# Peer review of "Ultrasonographic Evaluation of the Shoulders and Its Associations with Shoulder Pain, Age, and Swim Training in Masters Swimmers"

_medicina, 2020, doi:10.3390/medicina57010029_

Round 1

Reviewer 1 Report

Abstract

A word is missing in that sentence:The purpose of this study was to investigate the

prevalence of ultrasonographic abnormalities the shoulders in masters swimmers, and the

association of pain, age, and swim-training with structural changes in this population

Introduction: ok

Material and Method:

2.1 Participant: could you add the inclusion / exclusion criterias more clearly for each group

How did you recruit the participants? Especially the control one.

2.2 did you use any validated PRO (patient reported outcome) focused on shoulder ?

If not why

Discussion: could you add more clinical relevance or application

Author Response

Dear Reviewer 1,

Thank you for inviting us to submit a revision of our manuscript.

We appreciate the time and effort that you have dedicated in providing insightful feedback on our manuscript.

Based on your suggestions, we have incorporated changes into the manuscript and now hope that this manuscript addresses all previous concerns that were noted.

The attached PDF file is the manuscript with the corrections you have pointed out and highlighted in yellow lines.

Again, thank you for your kind review. We hope that these revisions persuade you to accept our submission.

Sincerely,

Authors

Reviewer 2 Report

This is an interesting u/s study regarding shoulder tendon alterations in master swimmers and comparison to healthy controls associating these changes with shoulder pain.

Methodology is sound, results are comprehensive and make sense and though the outcomes are something that expected and already known (except from SSC which was newly tested), I couldnot identify any flaws or mistakes.

I expect that the interest to the readership will not be massive as these changes are self-implied in swimmers. However I must recognize that it adds evidence to the already known and establishes additional data. 

I should however anticipate authors to have narrowed ages more strictly as one cannot compare elite swimmers of 40 years to 60 years-degeneration is a confounding factor-huge indeed. This limitation is mentioned by authors in discussion and this covers me.

Author Response

Thank you for the time and effort that you have dedicated in providing insightful feedback on our manuscript.

We are very happy to receive such a comment from you.

Swimming is not only a competitive sport, but also a lifelong sport, and swimming is one of the most popular sports for masters age group. Many swimmers take up swimming to improve their health. However, the swimmer's health should not be harmed (inducing shoulder pain) by swimming for improve his or her health. From this perspective, we started this study. Until now, there have been few studies on shoulder pain in the masters age group.

As shown in this study, shoulder pain in masters swimmers is caused not only by swimming training but also by aging.

In the future, we would like to collect larger-scale data to clarify the causes of shoulder pain in masters swimmers in more detail by grouping them according to their competition history, competition level, and training volume.

Thank you very much for your kind review.

Sincerely,

Authors